# Developing System-Oriented Interventions and Policies to Reduce Car Dependency for Improved Population Health in Belfast: Study Protocol

Ruth F. Hunter [1,*] , Claire L. Cleland [1] , Frank Kee [1], Alberto Longo [2], Brendan Murtagh [3], John Barry [4] , Gary McKeown [5] and Leandro Garcia [1,*]

1   Centre for Public Health, Queen's University Belfast, Belfast BT7 1NN, UK; c.cleland@qub.ac.uk (C.L.C.); f.kee@qub.ac.uk (F.K.)
2   School of Biological Sciences, Queen's University Belfast, Belfast BT7 1NN, UK; a.longo@qub.ac.uk
3   School of Natural and Built Environment, Queen's University Belfast, Belfast BT7 1NN, UK; b.murtagh@qub.ac.uk
4   School of History, Anthropology, Philosophy and Politics, Queen's University Belfast, Belfast BT7 1NN, UK; j.barry@qub.ac.uk
5   School of Behavioural Sciences, Queen's University Belfast, Belfast BT7 1NN, UK; g.mckeown@qub.ac.uk
*   Correspondence: ruth.hunter@qub.ac.uk (R.F.H.); l.garcia@qub.ac.uk (L.G.)

**Abstract:** Reducing car dependency requires orchestrated multi-sectoral, multi-policy action in a complex landscape. Thus, development of proposed interventions to reduce car dependency should be informed by systems thinking, complexity science, and socio-technical transition theory. We aim to co-design sustainable systems-oriented intervention approaches to reduce car dependency in Belfast. The study includes seven integrated tasks—1: Map stakeholders and partnerships influencing car dependency using stakeholder network analysis; 2: A review of systematic reviews regarding interventions to reduce car dependency; 3: Map-related policies via analysis of policy documents and semi-structured interviews; 4: A participatory group model building workshop to co-produce a shared understanding of the complex system perpetuating car dependency and a transition vision; 5: Using Discrete Choice Experiments, survey road users to evaluate the importance of transport infrastructure attributes on car dependency and on alternative modes of travel; 6: Citizen juries will 'sense-check' possible actions; and, 7: Stakeholders will interpret the findings, plan orchestrated multi-sectoral action, and agree on ways to sustain collaborations towards the common vision of reducing car dependency. We expect to attain a systemic view of the car dependency issue, potential intervention approaches to reduce it, and a framework for their integration through the co-ordination of stakeholder actions.

**Keywords:** systems science; car dependency; public health; complexity

## 1. Introduction

Car dependency is defined as high levels of per capita car travel, car-oriented land use patterns, and reduced transport alternatives [1]. For some, there is no other choice. For others, alternative options exist, but they are either far less attractive (e.g., the high cost, over-crowding, poor reliability, and expense of public transport; poor infrastructure and road safety issues for cyclists and pedestrians; the relative inconvenience to meet families' daily schedules) or based on embedded travel habits, convenience, or other behavioural drivers [1,2]. Northern Ireland is one of the most car-dependent regions in Europe [3]. An average person in Northern Ireland makes 82% of all their journeys by car, compared to 63% in the United Kingdom (UK), and just over 50% in the Republic of Ireland [3]. Despite various government targets over the last two decades, the 'modal shift' from car dependency to other modes of travel has not happened, with research showing that individuals reported to be 'more reliant' on their vehicle than in the previous 12 months,

the highest proportion in the past 5 years [3,4]. Furthermore, Belfast is one of the most congested cities in the UK, largely because too many people are driving too often. Public transport is expensive, comparatively unreliable, and inflexible (focusing on key commuter corridors), deepening the transport poverty of the most excluded communities [5]. Indeed, the use of walking, cycling, and public transport for all journeys remains low at 24% and has not changed between 2009 and 2019 [6].

Car dependency is clearly a public health issue but also creates significant economic externalities, environmental blight, and reinforces inequalities, in particular, differential access to services and facilities (see Table 1). The consequences for population health include lower levels of walking and cycling, inferior air quality, and higher noise pollution—all of which are associated with detrimental impacts on mortality, non-communicable disease (NCD), and mental well-being, as well as increased rates of road traffic injuries and deaths [7–9]. Reducing car dependency has the potential not only to ameliorate public health risk factors but could also contribute towards meeting the UK Clean Air Strategy air quality levels [10], the Belfast Climate Action Plan, the Belfast Climate Commission's Net Zero Carbon Roadmap, and several Sustainable Development Goals [11] (3. Good health and wellbeing; 9. Industries, innovation, and infrastructure; 10. Reduced inequalities; 11. Sustainable communities and cities; 13. Climate change; and 15. Life on land), the UK governments Net Zero Carbon Emissions by 2050 target, and the Paris Agreement (https://unfccc.int/process-and-meetings/the-paris-agreement/the-paris-agreement, accessed on 22 July 2021).

However, none of these can be achieved through a single intervention but require orchestrated multi-sectoral, interdisciplinary-based, and multi-policy action across a complex landscape, involving various actors with their own perspectives, mandates, resources, and agendas [12]. This also necessarily involves an explicitly complex systems-based analysis (i.e., seeing car dependency as part of the transportation/mobility system, which is also connected to other systems such as health, urban governance, planning/spatial development, energy, etc.). Thus, the development and testing of proposed interventions and policies to reduce car dependency should be informed by systems thinking, complexity science, and socio-technical transition theory. This assumes that car dependency is shaped at multiple levels: at the level of the individual (micro level) as well as by social norms (meso level) and the socio-economic, urban, capital, and political environments (macro level) [2]. Therefore, it is imperative to understand influences at each of these levels, as well as the dynamic interrelationships among them, to devise and sustain effective interventions to reduce car dependency.

Car dependency is a multi-level, multi-sectoral "wicked" problem [13]. Many of the most pressing public health hazards operate outside the boundaries of the health sector. The question is not car or public transport or cycling, but what mix and balance makes more sense from a 'full spectrum, whole costs and benefits' perspective. To date, public health interventions have largely focused on promoting walking and cycling with limited effect [14–16] and without addressing the root causes of car dependency. Many UK cities are debating how to reduce journeys by car, for example, through clean air zones, workplace parking levies, improving public transport, and creating dedicated networks for walking and cycling. Transport research has tended to focus on environmental infrastructure (macro level) [17], while psychological research has focused on individual agency (micro level) [18]. On the other hand, recent research underpinned by social practice theory (meso level) has focused on the utility of the trips/journeys themselves as the key drivers of modal choice [2]. More likely, all these levels, and the dynamic relationships among them, play an important role in shaping and perpetuating car dependency and, therefore, should be taken into consideration concomitantly.

We argue that solving car dependency requires a complex systems approach. Numerous political, economic, environmental, interpersonal, and individual factors dynamically interact to shape and sustain car dependency [19,20]. They usually work through complex causal pathways leading to a range of potential intended and unintended outcomes (the

latter ones are also generated by actions to address other seemingly unrelated societal issues, such as housing provision), and their full effects may take a long time to emerge and change [21]. However, policymaking has relied on or defaulted to dominant linear models, which are sub-optimal to deal with complex dynamics. Indeed, multi-level dynamic interactions and emergent phenomena within complex causal pathways can generate multiple divergent outcomes from a single system perturbation or intervention [22]. For these reasons, trying to set up traditional experimental designs to isolate the effects of a systemic intervention is well-nigh impossible. By applying a complex system approach, we can develop a suite of systems-oriented interventions that account for and leverage interactions across multiple levels and sectors for positive change. Moreover, systems thinking methods are well positioned to assist stakeholders in creating a consensual understanding of the problem and thereby co-devise optimum, co-ordinated strategies and interventions to address car dependency and improve population health [23]. Furthermore, the socio-technical transitions framework takes into account the complexity, ambiguity, contestation and uncertainty, and the need to incorporate multiple stakeholder perspectives and different knowledge bases and disciplines upon which to base policy, including innovative and 'disruptive' solutions. This framework sees the complex system being 're-ordered' via transitions processes.

**Table 1.** Summary of the net negative balance of a city dominated by private car transport.

| Domain | Evidence |
|---|---|
| Health | Compared to other areas in the UK, Northern Ireland has the highest number of road deaths by region [3] |
|  | Health impacts of air and noise pollution, mostly in terms of NCD and poor mental health |
|  | Reduced physical activity (walking and cycling) levels |
|  | Increased burden to the health care system with substantial medical costs |
| Society | People from more deprived backgrounds are less likely to own or have access to a car |
|  | When other transport options do not exist, it can reduce access to key destinations and services, therefore, widening inequality [24] |
| Environment | 23% of greenhouse gas emissions in Northern Ireland are due to transport [3] |
|  | Cars require a large amount of urban space (i.e., roads and car parking), meaning less shared spaces, parks etc. that have significant health-enhancing benefits |
| Economy | A weakening of key aspects of the local economy and city-based retail, and the generation of urban sprawl, with the latter often viewed as a misallocation of economic and other resources |
|  | There is an unacknowledged economic cost to a car-based mobility system where, for up to 80% of the time, the car is not in use (i.e., parked) [3] |

Therefore, our overarching aim is to co-develop sustainable and scalable systems-oriented interventions at the intersection of policy and environmental infrastructure, social context and environment, and individual agency to reduce car dependency for improved population health. Our focus will be on the Belfast City Region, a typical example of a UK city where car use levels are high and congestion (and related consequences) are increasing. Our objectives are to:

1.  Understand the structure and characteristics of the network of stakeholders influencing car dependency (e.g., investment in roads, spatial planning) (Task 1), and the development, implementation, and evaluation of interventions to create and sustain a transition to reduced car dependency (Task 2 and 3).

2. Develop consensus about the nature, ordering, and relationships among interventions from different sectors that influence (both positively and negatively) car dependency by developing shared understandings of the causal loops which shape the complex systems influencing car dependency (Task 4).

3. Evaluate and rank the importance of individual-level influences on car dependency and alternative travel modes (including working from home, i.e., non-travel options for some purposes) (Task 5).

4. 'Sense-check' and 'socialise' promising intervention approaches with road users and the general public (Task 6).

5. Co-develop and implement a framework for collaborative knowledge generation and synthesis, and explore the public and political acceptability, utility, affordability, feasibility, and sustainability of transitions (interventions, policies, actions, programmes, initiatives, and regulatory frames) to reduce car dependency and improve population health (Task 7).

## 2. Materials and Methods

The study is organised upon three pillars: (1) understanding the multiple layers that shape car dependency in Belfast (Objectives 1 and 2; Tasks 1 and 4); (2) synthesizing evidence and knowledge of what has worked in other cities (Objectives 1 and 3; Tasks 2, 3, and 5); (3) developing a shared vision and co-ordinate innovative actions with stakeholders to reduce car use in Belfast (Objectives 4 and 5; Tasks 6 and 7). The leadership of this study is shared by stakeholders from government, local authority, charitable and industry sectors, and researchers from different disciplines (e.g., public health, complexity science, urban planning, psychology, political science, and economics) to maximise shared learning and optimise decision making. Car users are involved in several of the tasks in the project to ensure that their insights are considered throughout the research. Our approach draws on the principles of the INDEX study, which provides guidance on how to develop complex interventions, in so far as it is dynamic, iterative, and creative, open to change and looking towards further evolution and evaluation. We also broadly follow the steps laid out by the INDEX framework [25], including: (1) involving stakeholders; (2) reviewing the evidence and theories; (3) primary data collection; (4) understanding the context; (5) attention to future implementation; and (6) design and refinement. We are also cognisant of the work of Moore et al. [26] through consideration of how the UK Medical Research Council (MRC) framework may be adapted for complex social system interventions leading to transitional outcomes for car use and the exercise of individual agency in, and increasing public support and legitimacy for, broadening multimodal transport choices.

Our programme of work has seven research tasks designed to enable us, in collaboration with stakeholders, to understand the underlying system, gather and synthesise the necessary evidence for action, and co-develop the interventions and policies. Figure 1 illustrates the integration of the tasks.

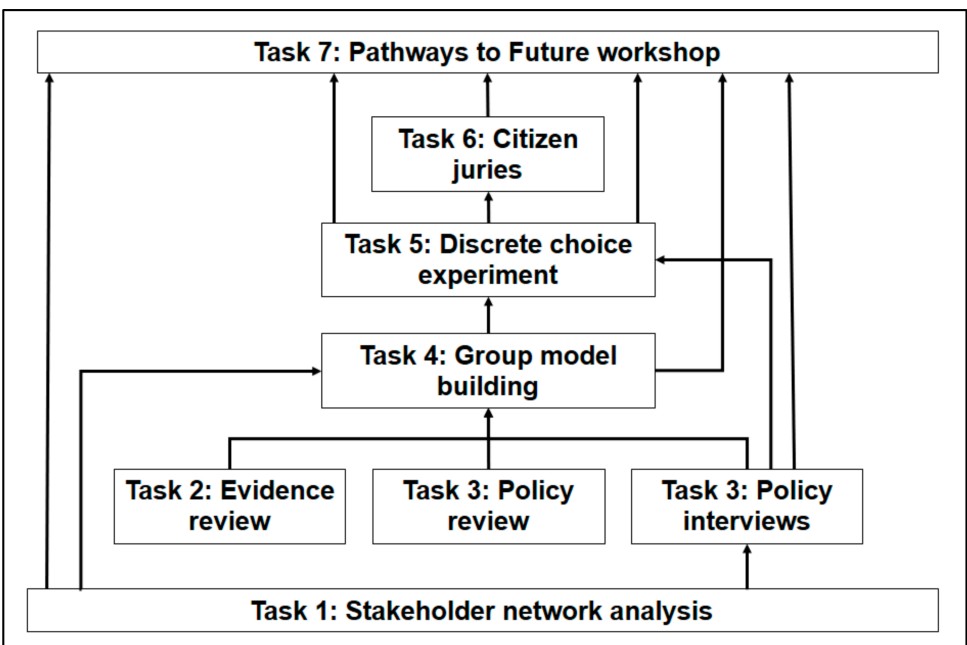

**Figure 1.** Schematic diagram of task integration. Tasks include: Task 1: identify and map the stakeholders influencing car dependency in Belfast and their relationships; Task 2: review the literature to learn from interventions and policies to reduce dependency; Task 3: map the policy landscape and plausible policy scenarios affecting car dependency in the Belfast region; Task 4: build mutual understanding of the system influencing car dependency in Belfast and potential co-ordinated multi-sectoral solutions; Tasks 5 and 6: understand the preferences and perspectives of car users in relation to a series of policy alternatives to reduce car trips; Task 7: agree with stakeholders on a future pathway/way forward to address car dependency in the region.

The beneficiaries of our project are: (1) Car users for work and school travel, leisure, shopping, and socialisation in Belfast, as we envisage the main actions will focus on this population; and (2) Other road users (e.g., public transport users, pedestrians, and cyclists) and the general public, as they will also directly benefit from a transportation system that is less reliant on cars and more sustainable and healthier, making them co-beneficiaries of cleaner air and less noise.

### 2.1. Task 1: Stakeholder Network Analysis

We will map the key stakeholders that influence the current transport system characterised by high levels of car dependency in Belfast. The organisational capacity and the effectiveness of these partnerships to meet their objectives will be assessed using stakeholder network analysis tools [27,28]. We will use a snowball sampling framework to recruit participants stopping when the saturation point has been reached. We will use the stakeholdernet.org online tool for data collection, collation, and analysis. The strength and extent of network ties among the stakeholders, in terms of reciprocal relationships, exchange of information, and resources (e.g., financial, physical, human), will be evaluated using a range of mathematical parameters describing the network [29]. We will explore the barriers and enablers for a range of organisations to either individually as a single stakeholder or collaboratively with others change the dominant transport system with the aim of reducing car dependency. Key outputs from this task include the identification of stakeholders with power, influence, resources, knowledge, and capacity to affect car dependency (either by seeking to continue/enable the current system or seeking to transform it) in Belfast and their relationships.

### 2.2. Task 2: Evidence Synthesis

In parallel with Task 1, we will undertake a review of systematic reviews [30] regarding effective and cost-effective intervention approaches and components to reduce car dependency at the population level. Given the transdisciplinary nature of the study, we will search a range of disciplinary electronic databases such as public health, transport, planning, policy, economics, and psychology. We will primarily focus on reviews published from 2000 onwards that have implemented a systematic methodological approach. Search terms under the following categories will be used: 'active travel', 'public transport', 'disincentivising car use', 'urban planning/design', and 'reviews'. One member of the research team will conduct the searches, who together with two other researchers will screen the abstracts and full texts and extract the data. Findings will be synthesised qualitatively and interpreted with our stakeholders. We will attempt to conduct a meta-analysis depending on the number and heterogeneity of the data extracted. Key outputs from this task include a review of the evidence base to support (or refute) current planned interventions to reduce car dependency and identification of new interventions to be considered by stakeholders. We will also develop an understanding of policy positions, regulatory determinants, interventions, and investments to develop a transitions framework in Task 3.

### 2.3. Task 3: Policy Mapping and Socio-Technical Transitions

Involving stakeholders identified in Task 1, we will map the ongoing and planned interventions that will likely affect car dependency within the Belfast transportation system and provide an understanding of their complex interplay. The modal shift from car dependency can be viewed as not about achieving a definable end state but as a process of redirecting and steering a wide range of factors (e.g., infrastructure, governance, individual behaviour, markets, energy, technologies) towards a more sustainable configuration. Using a 'multi-level perspective', this task will further develop the stakeholder analysis from Task 1 to explore the dynamics and relationships of the car transport system within Belfast at the landscape, system, and niche levels [31]. These levels include: (1) a micro-level of niches representing innovative local practices, local actors, and local technologies; (2) a meso-level system level relating to the dominant technologies, practices, policies, rules, shared assumptions, and discourses/norms; and (3) a macro-level landscape comprised of the social and physical environmental or infrastructural features within which the dominant system and niches are nested and subject to influence. We will undertake a critical analysis of the relevant policy documents from all government departments in Northern Ireland and conduct semi-structured interviews with 8–12 key stakeholders in the transportation system in Belfast to explore the main obstacles and opportunities. This analysis also recognises the varying degrees of power and influence, resources, legislative capacities, and how the dominant interest in roads planning (haulage, retail distribution, the car lobby, civil engineering professions, and so on) favour one form of modal transport over others. The key outputs from this task include a map identifying a transitions framework (regulations, policies, resources, and interventions) and their potential interplay in reducing car dependency. This provides the basis to co-create a shared understanding of the system influencing car dependency (in Task 4) and for citizen juries to determine practicable and sustainable interventions in reducing car dependency (in Task 6).

### 2.4. Task 4: Participatory Group Model Building (GMB)

Building on Task 3, the participatory GMB [32] will enable stakeholders from across sectors to consider different perspectives and build a shared understanding of the complex system influencing car dependency. We will conduct a GMB workshop with key local stakeholders representing multiple sectors, such as national- and local-level governmental agencies, road users, civil society and advocacy groups, private sector, and researchers. The workshop will be based on the Community-Based System Dynamics approach [23], which places emphasis on the development of capabilities to understand systems models and their uses, barriers, and limitations, especially in the delivery of a transitions framework

for reducing car dependency. A facilitation team will guide the activities and one observer will record the sessions, taking notes and documenting the discussions and key decisions. Three key outputs are expected from this task. First, a causal loop diagram depicting the consensual understanding of the system perpetuating car dependency in Belfast. This diagram will provide a synthesis of the stakeholders' multiple perspectives and the scientific and practice-informed evidence. Second, and together with Task 3, we will produce a shared transition vision and identification of pathways to reduce car dependency. Third, this task will produce a revised list of policy and intervention archetypes, building on the output of the policy mapping exercise (Task 3) and informed by the perspectives offered by the newly developed causal loop diagram and shared transition vision.

### 2.5. Task 5: Discrete Choice Experiments (DCE)

As we note, the costs and benefits of reducing car dependence are uneven and require a focused analysis on how the wider public might be incentivised to use alternatives to the car. Here, we will use stated preference methods (i.e., a DCE survey) [33] to: (1) explore how different communities experience the transport environment and use cars; (2) provide insights on the importance of various infrastructural attributes of the Belfast transport environment and on the agency of the individual; (3) understand the trade-offs that people are prepared to make among those different attributes; and (4) elicit people's willingness to pay for or willingness to accept changes in the provision of those attributes, according to policy options identified in Tasks 2 to 4. The DCE survey will target car users, including those who regularly travel into the Belfast city centre, to explore enablers and barriers in the demand for car dependency. An independent marketing research company will administer the DCE survey [34] to recruit 500 car users in Belfast, using a D-efficient sequential Bayesian experimental design [35,36] to provide a sample of the adult population of car users in Belfast representative of the population for age, gender, and socioeconomic characteristics. Data will be analysed using discrete choice models (e.g., multinomial logit and mixed logit models) to explore preference heterogeneity, both observed and unobserved, capture the differences across groups of individuals, and estimate predictions of people's preferred scenarios for the Belfast transport environment and their associated willingness to pay for or accept changes. The results from this task will be used within the citizen juries (Task 6) and Pathways to the Future (Task 7) to inform discussions.

### 2.6. Task 6: Citizen Juries

Building on tasks 2, 3, 4, and 5, we will undertake citizens' initiative review methods (i.e., citizen juries) [37], to elicit lessons and collective informed deliberations for action. This task draws the various empirical strands together to explore the most effective pathway to achieve the type of systemic (cultural, policy, behavioural, and environmental) transition of the existing transportation system through planning. We propose two citizen juries (n = 12–15 citizens per Jury) that will 'sense-check', socialise, and test the legitimacy, acceptability, utility, applicability, affordability, and feasibility of the possible intervention approaches identified in Tasks 2 to 5. These juries will reflect a mix of 'ideal types' to democratise and embed the evaluation approach such as: high dependency suburban car users; disadvantaged communities with low car access; women and young mothers; disabled and older people; and young people. This task will 'test' intervention approaches with citizens.

### 2.7. Task 7: Pathway to the Future

The empirical and conceptual work will be drawn together via a one-day workshop with the same participants of the GMB exercise (Task 4), with emphasis on: (1) collectively synthesising and interpreting the evidence from Tasks 1–6, accommodating different points of view, priorities, and sometimes contradictory evidence; (2) planning for an orchestration of ongoing and planned multi-sectoral actions, which together will form the intervention; and (3) agree on the best ways to sustain collaborations towards the

common goal of reducing car dependency in Belfast. We will guide the conversations in the workshop around the stages and principles of adaptive policy [38], designed to function more effectively in complex and dynamic conditions and more robust to uncertainties. According to this framework, the stages of the policy cycle are: policy set-up (done through Tasks 1–5), policy design and implementation, and monitoring and continuous learning. It is not realistic to expect that at the end of this workshop that the group of stakeholders will agree on all aspects related to policy design and implementation or monitoring. Rather, this workshop will serve to start and commit to a series of co-ordinated discussions to build synergistic actions towards the common goal of reducing car dependency. The key outputs of this task include a consensual understanding of the evidence provided in Tasks 1–6 and implications for reducing car dependency in Belfast. The conversations will orchestrate the co-ordination of ongoing and planned policies and actions, as a basis for agreement on mechanisms for collaborative work.

## 3. Expected Results

Our systems-oriented approach will provide the framework for the stakeholders to work in co-ordination to select, orchestrate, and co-adapt their initiatives to achieve a consensually defined common goal. To make progress requires a set of agreements and commitments among stakeholders: (1) a shared agreement that car dependency as a key component of the current Belfast transport/mobility system is a health, environmental, societal, and economic problem and must be reduced; (2) a mutual understanding of the system perpetuating car dependency in the region; (3) consensus that no single sector or initiative will solve the problem alone; (4) commitment to co-ordinate multi-sectoral, multi-level action towards reducing car dependency while maintaining or improving mobility options for citizens; and (5) a systems-oriented, co-produced approach that is deemed acceptable, feasible, and affordable by stakeholders and the citizen juries.

## 4. Discussion

Transforming transportation systems with the aim of reducing car dependency has the potential to mitigate a series of public health risk factors that contribute to the chronic disease burden in the UK. The key change being sought is in the transportation mode share, shifting from car reliance to more sustainable and healthier modes of travel, such as public transport, walking and/or cycling, and also reducing the need to travel in the first place for some purposes and activities. This will trigger change in a series of public health risk factors, namely: road traffic collisions, reducing injury and death rates, and injury severity; physical inactivity; air pollution (including carbon dioxide leading to climate breakdown); and noise pollution. The relationships among these factors sustain reciprocal and dynamic influences but with the potential to generate a virtuous cycle. For instance, fewer cars in the streets makes walking and cycling trips safer (fewer road collisions), encouraging more people to switch to these modes. By increasing the number of walkers and cyclists observed on the streets, social norms towards these modes are strengthened, encouraging more people to move from car travel to active travel modes and lessen road collisions further (as well as air and noise pollution). Our list of public health outcomes will incorporate the stakeholders' views (including unintended consequences) and impacts on health inequalities.

The exact nature of the systems-oriented interventions and policies will be informed by the research outlined above. Here, we speculate on what that may involve based on current contextual knowledge. A range of policies at various levels of the regulatory process in Belfast are currently being considered. These include: improved public transport provision, with the introduction of phase 2 of a rapid transit system [39]; investment in urban greenways [40]; expanding the car-free areas of the city core and lowering the speed limit in more part of Belfast; building capacity for sustainable travel, particularly cycling infrastructure [41]; extension of 'park and ride' facilities at the edge of the city; development of 30,000 new houses in the inner city area [42]; and improvements of public

open spaces that encourage active travel [42]. Other intervention options that may be explored include free public transport for all school children, tighter regulation of car parking, congestion charging, and social marketing campaigns to change social norms on car use [43]. An emerging issue due to the COVID-19 pandemic is how working from home will continue to be part of a new 'blended' work experience, reducing the amount of work-related travel for those working in sectors where it is possible and feasible to work from home. However, we must also be cognisant of the potential for other initiatives to subsidise car dependency, such as road building, road expansion, the increase in and reduction of the cost of car parking, and the political power of car-based interests such as road hauliers, car manufacturers, and car dealers. Such initiatives include the widening of a four-lane carriageway to a six-lane freeway, continued expansion of city centre parking (e.g., 900 spaces at Belfast Harbour), etc. In addition, we must also consider the significant contextual implications of the post-Brexit and post-COVID-19 policy agendas, and any future interventions and policies must be robust and adaptable to a somewhat uncertain future (societal, political, economic, etc.) landscape.

Each of these initiatives have positive and negative impacts, but without an integrated long-term vision, there could be conflicting and unintended consequences, suboptimal outcomes, and public as well as policy resistance.

## 5. Conclusions

In summary, numerous policies have been implemented in a bid to reduce car dependency with several European countries being at the forefront of active travel and public transport promotion [44]. Success in these countries (such as Germany, Switzerland, and Austria) has involved the implementation of multi-level (i.e., national, regional, local) innovative programmes that have sought to create a change in the complex transport system (e.g., public transport or active travel or road safety) by implementing more balanced transport systems and solutions [44].

However, this is not the case for all cities and countries [45]. For example, many transport policies lack geographical vision leaving those who reside beyond the boundary of a city with no other option but to use a car as their primary mode of transport. Other transport policies have failed to take account the land-use mix and/or the land-use density and have not provided high quality active travel infrastructure and public transport services [44]. Consequently, shifts in the car dominant model are unlikely to occur, and the car will continue to be prioritised as the dominant mode of transport until a time when all aspects of the complex transport system are considered and a systems-wide approach is taken.

At the end of the study, we expect to attain a systemic view of the car dependency issue in Belfast, a view of the potential policies and intervention approaches, and a framework for their integration and for the co-ordination of stakeholders.

**Author Contributions:** Conceptualization, R.F.H., L.G., F.K., J.B., A.L. and B.M.; methodology, R.F.H., L.G., F.K., J.B., A.L. and B.M.; formal analysis, R.F.H., L.G., C.L.C., F.K., J.B., G.M., A.L. and B.M.; investigation, R.F.H., L.G., C.L.C., F.K., J.B., G.M., A.L. and B.M.; data curation, R.F.H., L.G., C.L.C., F.K., J.B., G.M., A.L. and B.M.; writing—original draft preparation, R.F.H.; writing—review and editing, R.F.H., L.G., C.L.C., F.K., J.B., G.M., A.L. and B.M.; project administration, R.F.H. and L.G.; funding acquisition, R.F.H., L.G., F.K., J.B., G.M., A.L. and B.M. All authors have read and agreed to the published version of the manuscript.

**Funding:** The study is funded by a grant from the MRC Public Health Intervention Development (PHIND) scheme (MR/V00378X/1). We also acknowledge funding from the HSC Research and Development Office Northern Ireland. The funding body has no role in the design of the study and collection, analysis, and interpretation of data and in writing the manuscript.

**Institutional Review Board Statement:** The study will be conducted according to the guidelines of the Declaration of Helsinki, and approved by the Faculty of Health, Medicine and Life Sciences Ethics Committee, Queen's University Belfast (protocol code MHLS20_141 and 23 November 2020).

**Informed Consent Statement:** Informed consent will be obtained from all subjects involved in the study.

**Data Availability Statement:** The datasets used and/or analysed during the current study are available from the corresponding author on reasonable request.

**Acknowledgments:** We would like to acknowledge our study partners: Department for Finance Innovation Lab, Department for Health, Department for Infrastructure, Belfast Healthy Cities, Belfast City Council, Translink, and Sustrans.

**Conflicts of Interest:** The authors declare no conflict of interest.

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
