# Peer review of "Developing System-Oriented Interventions and Policies to Reduce Car Dependency for Improved Population Health in Belfast: Study Protocol"

_systems, doi:10.3390/systems9030062_

Round 1
Reviewer 1 Report
General comments:
I get the idea behind the proposed research project, but I have a feeling that the authors are trying to do too many things. I struggled, in particular, to understand how the seven “tasks” they discuss build on each other in a coherent way. Some of them (e.g., group model building and “citizen juries”) seem redundant.
Here are my suggestions:
- Go ahead with the literature reviews. Can’t hurt, might help.
- Go ahead with the stakeholder identification.
- Go directly to group model building. I’m not sure if you’re going to do only Causal Loop Diagrams or go all the way to a stock-and-flow model. My guess is that many of your participants will benefit from the CLD but will get less benefit from a true quantitative model, unless you can make it more like a simulation/video game.
- Forget the citizen juries as a separate task. The participants in your group model building exercise are your citizen “jurors,” especially the car users that will be part of that workshop.
- Make sure to involve all your workshop participants in the final task of identifying potent policy choices.
My final advice is to be realistic about what parts of your policy mix will get implemented, and how effective they will be. My guess is that the authorities in Belfast have already tried some things, with little apparent success. Perhaps your project will go deeper in identifying the systemic elements that have created this stubborn chronic problem, but it may not. It’s worth a try, though.
Detailed comments:
Lines 38-39: Interesting definition, especially the last three words.
Lines 56-57: Good that you explained why car dependency is a public health issue. I didn't initially think it was all that clear.
Line 95: Should be "among them" (since there are more than two interrelated factors). The authors do this several other times in the manuscript: lines 160, 198, and 291.
Line 127: Interesting set of objectives.
Line 162: Should be "fewer.” The authors did this twice on this line.
Line 195: I know what policy resistance is, but I don't understand how the authors think it differs from public resistance. I mean, policy resistance comes from the behavior of the "public."
Lines 211-213: I don't see "car users" on your list. It seems obvious to me that you should have a robust number of actual car users involved in this project. Or are you planning to get their input from surveys? My experience with this type of thing is that users will respond one way on a survey but then act completely differently once changes are put in place.
Reading further, I see that you include them in group model building, surveys and in “juries.” But I still think it might be worth including them earlier.
Line 215: I have no idea what the INDEX study is. My guess is that many, perhaps most, of your readers will also be ignorant of it. So I recommend that you give a full summary of it.
Line 230 (Figure 1 caption): It might help the reader if the authors somehow connected the numbered items in this caption with the graphic/text items depicted on the diagram. The correspondence is not intuitive.
Line 322: I'm struggling to understand how each of these tasks builds on the other. On this task, I am unclear on what kind of model the workshop participants will work. Will the facilitator help in the development of a causal loop diagram and, later, a stock-and-flow system dynamics model? If so, how will be previous tasks contribute? Why are they needed if, in this step, a CLD and SD model will be developed from scratch anyway?
Line 343: Wouldn't the various participants in the group model building workshop already have done the elements of this task? This seems redundant to me....
Reviewer 2 Report
Dear authors:
I have allowed myself to thoroughly review your manuscript, the topic itself, is very interesting, but I consider that the structure and presentation of results should better.
I attach my observations.
|
Line |
COMMENT |
|
54 |
Please include a paragraph mentioning the current status of policies created to reduce car use (e.g. Netherlands) or attempts and why these have not been successful. |
|
127-149 |
Objectives should be stated in a final paragraph in a continuous form. (not "the objectives are" and do not use bullet points). |
|
--- |
I do not understand the structure in which you present your manuscript, (Introduction, discussion, Materials and methods, and conclusions. |
|
--- |
The results are not clearly evidenced, do you mix it up with materials and methods? |
|
207 |
The methodology must be clear, explain how it will respond to the established objectives, clearly set two, at most three objectives to be analyzed. |
|
--- |
for a better understanding of the results I recommend that you put subtitles according to the established objectives, i.e. that the results respond to the objectives set. |
|
377 |
The conclusions are very brief. They must be derived from each objective and its results, it is a chain.
It is possible that they got the wrong version of the manuscript. |
|
|
You should include at least 50 references to strengthen your manuscript. |
Reviewer 3 Report
The Authors aim to co-design sustainable systems-oriented intervention approaches to reduce car dependency in Belfast. The topic is well developed and potentially interesting for the readers, although it would have been preferable to have some preliminary results. I have some comments:
- what are the main difficulties that can be found in each of the 7 tasks, briefly describe possible solutions and alternatives;
- for each task, make a brief preliminary analysis of the costs and give an idea of ​​how you intend to cover them;
- divide the 5 study objectives into primary and secondary, and make a hypothesis on the timing of each objective.
Round 2
Reviewer 1 Report
While I still somewhat disagree with some of the authors' decisions on methods, I think they've made adequate arguments for the approaches they took. I also think they've cleaned up a few minor English glitches and substantially clarified other arguments that I earlier thought were unclear.
Author Response
Thank you for your careful review of our manuscript.
Reviewer 2 Report
I have read again the whole document and it is very complex to try to understand the idea, I think it is because of the structural form (not of the presentation, but of the project). The authors state that it is the structure of the journal. In this journal it will be the first manuscript (study protocol) to be published, if accepted. I suggest you review the structure of other MDPI manuscripts and you will notice the structure and presentation of information.
https://www.mdpi.com/1660-4601/18/12/6187/htm
https://www.mdpi.com/1660-4601/18/11/5713/htm
https://www.mdpi.com/2227-9032/9/5/600/htm
https://www.mdpi.com/2673-527X/1/3/18/htm
When I mentioned about the structure and that it is not understood, I mean that it is not evident where the study is going, call it "expected results" if you wish. Your study does not have a correct structure. If you want to add this title "expected results" and connect it to your objectives and methodology, and maybe they will just realize if the design is correct. Or, on the other hand, include somewhere in your manuscript where you will get to with this one.
Additional:
- The literature review must be strengthened. Do not be afraid to cite more in the introduction.
- Think of a technique called "expert judgment" will greatly simplify your methodological process and will be more understandable your study or future study (so far it is not clear to me). Eliminate citizen jury, they will surely have representatives, take them to the expert judgment.
- Who really are the stakeholders, the specific objectives are not clear.
- Your methodology focuses on three "pillars" leaving out two pillars stated by you.
- Discussion before Methodology, I know that other journals can handle it this way, but here I would not think so.
- Do you really expect "At the end of the study, we hope to achieve a systemic view of automobile dependence" or do you expect: To develop interventions and policies...
